# Alternative oxidase causes cell type- and tissue-specific responses in mutator mice

Lilli Ikonen[1], Sini Pirnes-Karhu[2], Swagat Pradhan[1], Howard T Jacobs[3], Marten Szibor[3,4], Anu Suomalainen[1,5]

Energetic insufficiency, excess production of reactive oxygen species (ROS), and aberrant signaling partially account for the diverse pathology of mitochondrial diseases. Whether interventions affecting ROS, a regulator of stem cell pools, could modify somatic stem cell homeostasis remains unknown. Previous data from mitochondrial DNA mutator mice showed that increased ROS leads to oxidative damage in erythroid progenitors, causing lifespan-limiting anemia. Also unclear is how ROS-targeted interventions affect terminally differentiated tissues. Here, we set out to test in mitochondrial DNA mutator mice how ubiquitous expression of the *Ciona intestinalis* alternative oxidase (AOX), which attenuates ROS production, affects murine stem cell pools. We found that AOX does not affect neural stem cells but delays the progression of mutator-driven anemia. Furthermore, when combined with the mutator, AOX potentiates mitochondrial stress and inflammatory responses in skeletal muscle. These differential cell type-specific findings demonstrate that AOX expression is not a global panacea for curing mitochondrial dysfunction. ROS attenuation must be carefully studied regarding specific underlying defects before AOX can be safely used in therapy.

## Introduction

Genetic mutations that disrupt mitochondrial gene expression can lead to the defective assembly of respiratory chain (RC) complexes and appear to be a frequent cause of mitochondrial diseases. Myopathies, muscle diseases, are the most common mitochondrial disease phenotype in adults ([1], [2], [3]). Exactly how mitochondrial dysfunction, particularly RC dysfunction, leads to highly variable mitochondrial disease phenotypes is still poorly understood. Expanding this knowledge further is necessary to develop treatment options for these currently incurable diseases.

"Mutator" mice are a commonly used model to study mitochondrial dysfunction ([4], [5]). They carry a knock-in inactivating mutation in the exonuclease domain of the catalytic subunit of mitochondrial DNA (mtDNA) polymerase gamma (POLG; p. D257A), leading to mtDNA mutation accumulation and increased mtDNA replication ([6], [7]). These irregularities cause various cell type-specific defects, especially affecting stem cell pools via increased reactive oxygen species (ROS)-related signaling ([8], [9], [10]). Furthermore, they show abnormal cell cycle progression and nuclear genomic DNA breakage in stem cells because of imbalanced nucleotide pools ([7]). These defects lead to a progeroid phenotype starting from 6–8 mo of age with progressive hair graying, hair loss, osteoporosis, general wasting, and decreased fertility ([4], [5]). The lifespan of mutator mice is limited to 13–15 mo by severe anemia, which develops alongside a decline in lymphopoiesis ([11]). In addition, postmitotic tissues demonstrate mildly progressive RC dysfunction, causing mild mitochondrial myopathy in skeletal muscle and cardiomyopathy ([4], [5], [12], [13]).

The excess production of ROS partially causes the mutator stem cell homeostatic defect. In adult mutators, bone marrow shows highly increased oxidative stress within mitochondria, delayed mitochondrial exclusion from red blood cell (RBC) precursors, and extended iron loading by transferrin, leading to excess ROS generation via the Fenton reaction and oxidative damage in erythrocyte membranes ([9]). In other tissues, signs of ROS-related damage have not been found ([4], [5]). The antioxidant and reducing agent N-acetyl-L-cysteine (NAC) improves aberrant ROS signaling, rescuing fetal neural stem cell stemness and hematopoietic progenitor differentiation. In contrast, mitochondrially targeted ubiquinone, MitoQ, improves erythroid differentiation but is highly toxic to neural stem cells ([8], [10]). These findings indicate differential sensitivities of stem cell compartments to ROS-modifying treatments and highlight the importance of in vivo studies to collect evidence of cell type-specific effects of mitochondrial-targeted interventions.

Alternative oxidase (AOX) resides in the inner mitochondrial membrane as part of the RC (Fig 1A) and is present in most eukaryotes, including metazoan taxa, but absent from insects and mammals. Under specific stress conditions, AOX expression

---

[1]Stem Cells and Metabolism Research Program, Faculty of Medicine, University of Helsinki, Helsinki, Finland   [2]Research Program for Clinical and Molecular Metabolism, Faculty of Medicine, University of Helsinki, Helsinki, Finland   [3]Faculty of Medicine and Health Technology, Tampere University, Tampere, Finland   [4]Department of Cardiothoracic Surgery, Center for Sepsis Control and Care, Jena University Hospital, Friedrich-Schiller University of Jena, Jena, Germany   [5]Helsinki University Hospital, HUSLAB, Helsinki, Finland

Correspondence: lilli.ikonen@helsinki.fi; anu.wartiovaara@helsinki.fi

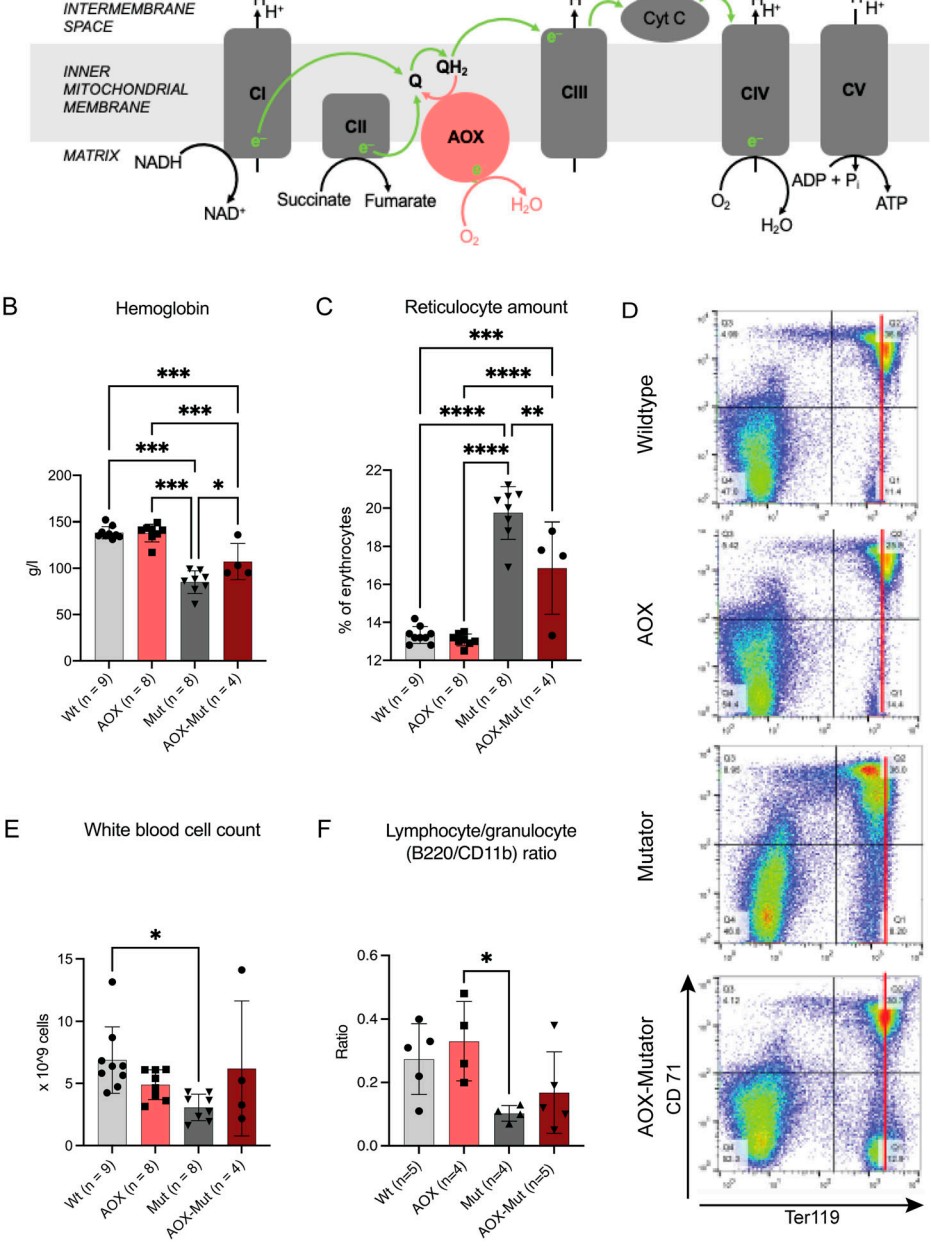

**Figure 1. Alternative oxidase (AOX) expression alleviates anemia in 10-mo-old mutators by shifting erythroid precursors toward a more mature state.**
**(A)** A simplified scheme of the respiratory chain demonstrating the capacity of the AOX to bypass functions of complexes III (CIII) and IV (CIV). **(B, C)** Hemoglobin and reticulocyte amount from mouse blood. **(D)** Hematopoietic precursor analysis from bone marrow. Erythrocyte FACS dot plots for Ter119 (maturing forms) and CD71 (early precursors). **(E)** White blood cell count from mouse blood. **(F)** Lymphocyte/granulocyte ratio calculated from FACS data. Samples are biological replicates in the numbers presented in the figure; each sample was analyzed once. All graphs are mean with SD. Statistical significance determined using one-way ANOVA with $P$-values: * ($P \leq 0.05$), ** ($P \leq 0.01$), *** ($P \leq 0.001$), and **** ($P \leq 0.0001$). Abbreviations: CI–V, respiratory chain complexes; NADH, NAD$^+$, reduced and oxidized forms of nicotinamide adenine dinucleotide; QH$_2$, Q, reduced and oxidized forms of ubiquinone; WT, wildtype mice; AOX, AOX mice; AOX–mut, AOX–mutator mice; mut, mutator mice.

mitigates over-reduction of the quinone pool, the consequent production of excess ROS, and the accumulation of keto acids (14, 15). In addition, AOX prevents the conversion of excess reducing power accumulated in the ubiquinone pool to superoxide. It does this by reducing O$_2$ to H$_2$O, bypassing the enzymes ubiquinol: cytochrome c oxidoreductase and cytochrome c oxidase (COX), that is, complexes III and IV of the respiratory chain, respectively, and decreasing mitochondrial membrane potential and ATP synthesis (Fig 1A). This relaxes the coupling of electron transfer to ATP synthesis and decreases ROS production and reductive stress (Fig 1A). AOX from *Ciona intestinalis*, a sea squirt, has been xenotopically expressed in mammalian cells, flies, and mice (14, 15, 16, 17). As proof

of principle, AOX rescued fruit flies from death upon exposure to cyanide, an inhibitor of complex IV. AOX expression has been reported to be innocuous in human cell culture, *Drosophila*, and mice, except for one study where AOX expression in a COX15 knockout mouse model worsened the myopathy and shortened the lifespan of these animals (14, 16, 18, 19, 20, 21, 22, 23, 24). AOX also compensates for respiratory chain dysfunction, particularly when complex III or IV activity is limiting (14, 16, 18, 20), and restores respiration linked to complexes I and II, preventing lethal mito-chondrial cardiomyopathy in a mouse model (25). Different mouse tissues, including skeletal and heart muscles and blood cells, widely express AOX when driven by the synthetic CAG promoter (17).

However, it is not known whether AOX is expressed in somatic stem cells and whether its effect on redox reactions affects stem cell homeostasis.

In this study, we used the combination of AOX and mutator mice to determine whether and eventually how AOX expression affects (a) somatic stem cell function in healthy mice, (b) stem cells subjected to mtDNA mutagenesis, and (c) respiratory chain function in key postmitotic tissues including skeletal muscle. We report that AOX expression has tissue-specific consequences in mutator mice. It alleviates anemia in mutators by shifting erythroid precursors toward a more mature state in bone marrow, whereas also inducing the mitochondrial integrated stress response (ISRmt) and inflammatory pathways in skeletal muscle. Our findings indicate that AOX expression may be helpful in hematopoietic stem cells experiencing mitochondrial stress but potentially harmful in postmitotic tissues via the induction or exacerbation of metabolic stress and inflammation.

# Results and Discussion

### Generation of AOX–mutator mice

The mouse strains used in the study, that is, those expressing *C. intestinalis* AOX (AOX mice) and those with a knock-in mutation (p.D257A) inactivating the proof-reading exonuclease domain of DNA polymerase gamma (POLG; mutator mice) have been described previously in references 4, 5, 17, respectively. The strains were maintained in the C57Bl6/JOlaHsd background, with AOX mice as hemizygotes and mutators as heterozygotes. The mutator allele was transferred only via the paternal line to prevent the accumulation of maternally inherited mtDNA mutations. We obtained double-transgenic mice expressing AOX that were also homozygous for the POLG (p. D257A) mutator mutation. AOX, mutator, AOX–mutator, and WT littermates were born approximately in the expected Mendelian ratios (Fig S1).

AOX–mutator double-transgenic mice's gross phenotype and outward appearance were indistinguishable from that of mutator mice, with signs of progeria starting at 6 mo. Neural stem cells (NSCs) and postmitotic skeletal and heart muscles were confirmed to express AOX (Fig S2A–E).

### AOX expression alleviates defective hematopoiesis in mutators

Mutator mice manifest anemia and increased amounts of reticulocytes (4, 8, 11). Therefore, we asked whether the effect of AOX on redox metabolism affects hematopoietic progenitor differentiation in WT or mtDNA mutator mice. We harvested the bone marrow and peripheral blood of WT and mutator mice with and without AOX expression at 43 wk. At this age, mutators showed various progeroid signs such as kyphosis, alopecia, weight loss, osteoporosis, and anemia.

AOX–mutators showed higher hemoglobin (Hb) and lower circulating reticulocyte (immature RBC) counts than mutators (Fig 1B and C). AOX expression did not affect the number of erythroid precursors in adult mutator bone marrow (Fig S3A and B). However,

FACS analysis revealed that mutators show lower mean intensity of the signal for Ter119 and an absence of cells with selective high Ter119, a specific marker of mature erythrocytes (26). Furthermore, AOX–mutator bone marrow shows Ter119 signal with mean intensity resembling WT controls and the presence of cells with Ter119 high only, as seen in WT (Figs 1D and S3A and B). These results show that AOX expression shifts the maturation pattern of mutator erythroid precursors towards that seen in WT. Total white blood cell counts trended towards WT values in AOX–mutators compared with mutators (Fig 1E), as did bone marrow lymphocyte/granulocyte ratios (Fig 1F). However, these counts varied considerably in all groups (Fig 1E and F). We observe that AOX partially rescues the abnormal erythrocyte differentiation pattern seen in mutators (Figs 1D and S3A and B) similarly to NAC (8) and MitoQ (10), a modified ubiquinone targeted to accumulate in mitochondria (27). In conclusion, AOX affects erythrocyte differentiation in mutators but not in WT mice and partially rescues the anemia, which is ultimately fatal to mutator mice.

Because of the effect of AOX on mutator hematopoiesis, we proceeded to look at its effect on NSCs. Previously, we demonstrated that mutator NSCs show decreased "stemness," presenting a decreased ability to self-renew in clonal culture (8). AOX expression did not rescue defective self-renewal or proliferation of mutator NSCs (Fig S3C and D). Compared with erythrocyte precursors, mutator NSCs appear to be less sensitive to the effects of antioxidants, with only NAC treatment, but neither MitoQ nor AOX able to restore their self-renewal capacity (10). AOX expression did not affect the stemness or growth of WT NSCs (Fig S3C and D). Although AOX is expressed in most tissues, including NSCs (Fig S2A) (17), the differential expression of AOX in bone marrow or different blood-cell populations has not been studied and might explain the distinct effects on neural progenitors and hematopoietic cells.

These data demonstrate differential effects of AOX expression on different somatic stem cell compartments, but only under conditions of mitochondrial dysfunction. In WT mice, AOX expression did not affect erythroid, lymphatic or neural progenitors or their differentiation pattern.

### AOX expression affects respiratory chain components in mutator skeletal muscle

After examining the effect of AOX on stem- and progenitor-cell compartments, we asked whether it could provide a therapeutic benefit and alleviate RC dysfunction previously demonstrated in the postmitotic skeletal muscle of mutators (4, 8). We did not observe any significant differences between skeletal and heart muscles, so further analysis on heart muscle was not carried out. Histochemistry for complex IV (COX, partially encoded by mtDNA) and succinate dehydrogenase (SDH), complex II, nuclear-encoded revealed no COX-negative, SDH-positive fibers indicative of mitochondrial dysfunction in the skeletal muscle of mice of any of the genotypes analyzed (Fig 2A). However, the overall COX-staining intensity was decreased, compared with WT, both in mutators and AOX–mutators, consistent with diminished COX activity (Fig 2C). Only in mutators we observed a subset of COX muscle fibers with intense staining, from here referred to as "hyperpositive," a novel finding (Fig 2A and B). AOX–mutator skeletal muscle was lacking in

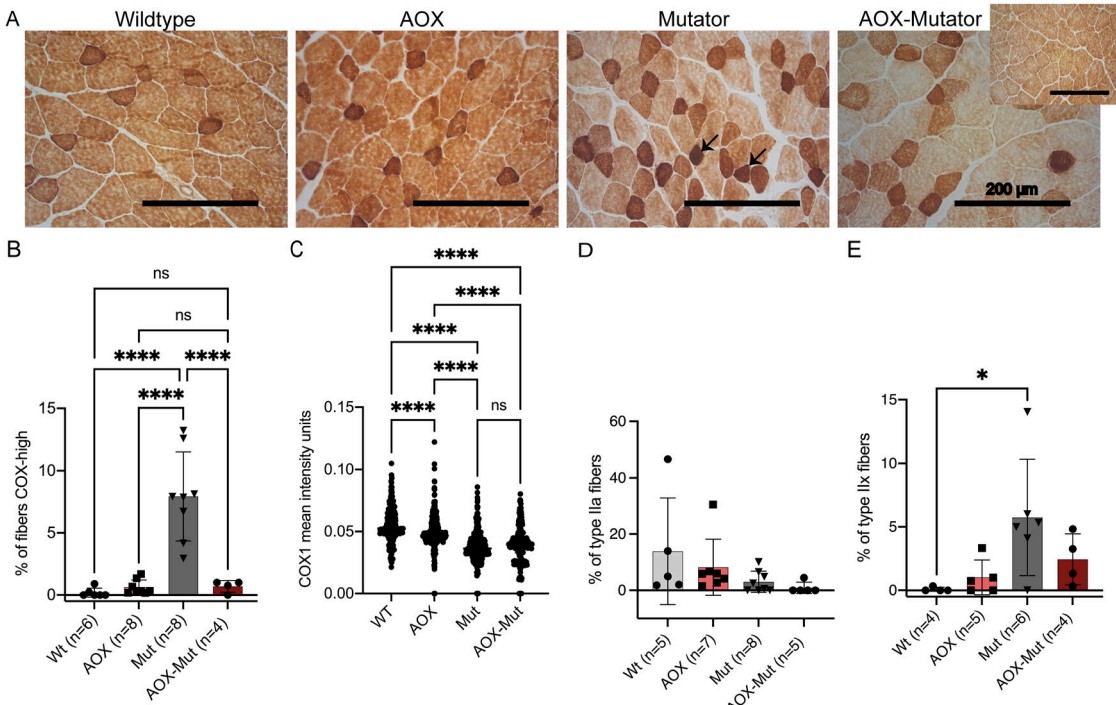

**Figure 2. Alternative oxidase (AOX) expression affects respiratory chain components in mutator skeletal muscle.**
**(A)** Cytochrome *c* oxidase (COX), brown precipitate and succinate dehydrogenase, blue precipitate; visible only in cells with COX deficiency histochemical activity assay from skeletal muscle of WT, AOX, mutator, and AOX–mutator mice. The small image for AOX–mutator shows homogeneous staining seen commonly in some sections. Arrows indicate example fibers with intense COX activity. Magnification 20x, scale bar 200 μm. **(B)** Quantification of fibers with intense COX activity; COX–succinate dehydrogenase histochemical activity assay on frozen skeletal muscle sections. **(B, C)** Scatterplot of COX1 mean intensity units; IF COX1 analysis of (B) in skeletal muscle. **(D, E)** Quantification of muscle fiber types, IF analysis of myosin heavy-chain type IIa (D) and type IIx (E) in skeletal muscle. Samples are biological replicates in the numbers presented in the figure; each sample was analyzed once. All graphs are mean with SD. Statistical significance determined using one-way ANOVA with *P*-values: * (*P* ≤ 0.05), ** (*P* ≤ 0.01), *** (*P* ≤ 0.001), and **** (*P* ≤ 0.0001).

these highly COX-positive fibers but had a similar overall COX-staining intensity (Fig 2A–C). Increased COX activity indicates increased mitochondrial activity; AOX is known to decrease mitochondrial biogenesis upon mitochondrial dysfunction in COX15-deficient mice (22). Therefore, the expression of AOX does not rescue COX protein amounts to the WT level (Figs 2C and S4A) but seems to impact the dysregulation of RC function in the muscle of mutators.

We proceeded to look at RC components at the protein level by immunofluorescent staining for the COX1 subunit, which was also decreased in mutators and not rescued by AOX (Figs 2C and S4A). AOX also did not affect COX1 expression in otherwise WT mice (Figs 2C and S4A). Immunohistochemistry for the SDHA subunit of the fully nuclear-coded complex II showed no statistically significant difference in intensity between the groups, but AOX mice appeared to have slightly lowered SDHA protein amounts (Fig S4B and C). No major mitochondrial mass changes as assessed by TOM20 were observed except for in AOX-WT mice that appeared to have slightly lowered TOM20 protein levels (Fig S4D). COX1 intensity correlates strongly with TOM20 intensity (Fig S4A and D) and SDHA intensity (Fig S4C) suggesting increased mitochondrial biogenesis in these fibers.

We also asked whether the alteration in COX activity and protein amount in mutators could be related to a shift in muscle-fiber type. We explored myosin heavy-chain isoforms preferentially associated with either oxidative or glycolytic metabolism in skeletal muscle. Based on immunohistochemistry for myosin heavy chain expression, which enables different muscle-fiber types to be distinguished (28, 29), mutators showed a decreased representation of smaller, oxidative type-IIa fibers compared with WT mice (Figs 2D and S4E), but this was unaffected by AOX expression (Figs 2D and S4E). Glycolytic type-IIb muscle fibers were prevalent in all mouse groups (Fig S4F). However, mutators showed an increase in type-IIx glycolytic fibers compared with the other groups (Figs 2E and S4G). There were no type-I oxidative fibers in the muscle of any mice (Fig S4H). The shift toward glycolytic muscle metabolism is consistent with decreased oxidative phosphorylation and COX activity, although the mitigation of this effect in the presence of AOX suggests that AOX may interfere with the stress signaling involved.

In deletor mice, a well-characterized model of mitochondrial myopathy, RC deficiency manifests in a mosaic pattern in skeletal muscle, alongside mosaicism for the promotion or prevention of mitophagy (30). Our finding of COX-hyperpositive fibers in mutator skeletal muscle further highlights the need for single-cell studies in the pathology of mitochondrial dysfunction, because a finding of decreased average COX levels would overlook both types of mosaicism.

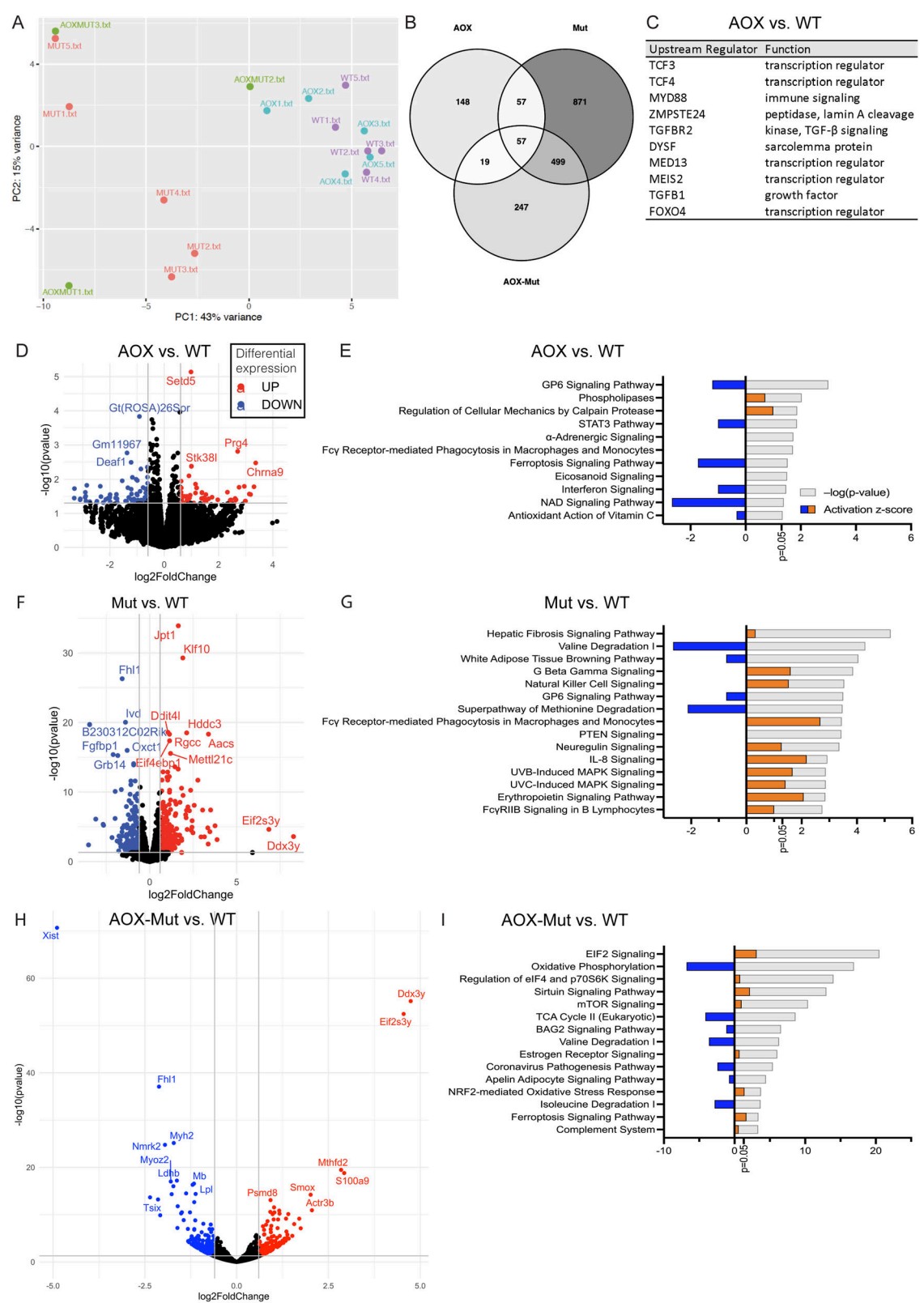

**Figure 3. Alternative oxidase (AOX) and mutator mice have distinct gene expression patterns in skeletal muscle.**
**(A)** Factors explaining variance in RNA sequencing (RNA-seq) of skeletal muscle samples analyzed using principal components analysis. **(B)** Overlap of significantly changed transcripts in RNA-seq in AOX, Mut, and AOX–-Mut compared with WT. **(C)** Top 10 activated upstream regulators in AOX compared with WT in RNA-seq data of mouse skeletal muscle analyzed using ingenuity pathway analysis. **(D, F, H)** Transcripts with the highest experimental fold change differing significantly between (D) AOX,

In the brain, aging mutator mice show diminished COX activity in multiple brain regions (cortex, hippocampus, accumbens, striatum, and thalamus) with a smaller decrease in the cerebral cortex (31, 32). In agreement with our previous results (8), we could not identify COX-negative, SDH-positive neurons in the hippocampus or dentate nucleus of mutators. However, RC-deficient cells were present around the lateral ventricles in the subventricular zone where NSCs reside. AOX expression did not alter the pattern of COX-SDH activity staining in the brain (Fig S5A and B).

### AOX expression in WT mice leads to altered genome methylation and decreased inflammation

In plants, AOX modifies mitochondrial signaling and affects nuclear transcription programs (33). To gain insight into the effects of AOX in mice, we performed RNA sequencing (RNA-seq) on mouse skeletal muscle with or without AOX in the WT and mutator backgrounds (Fig 3). Two principal components explained 58% of the variance between samples (Fig 3A). Mutators and AOX–mutators displayed the highest overlap in significantly altered transcripts (Fig 3B). When compared with WT mice, mutators and AOX–mutators had a substantially larger amount of significantly altered RNA transcripts than AOX mice (Fig 3D, F, and H). The number of significantly altered transcripts was similar in comparisons of AOX with WT mice and AOX–mutators to mutators (Figs 3H and 4A, Supplemental Data 1).

The histone methyltransferase Setd5 was the most significantly altered transcript in AOX versus WT (Fig 3D), whereas transcripts of many methylation-regulated genes (e.g., *Chrna9*, *Prg4*, *Zbp1*, and *Zdbf2*) were among those with the greatest fold change (Fig 3D, Table S1). Posttranslational modifications such as methylation are important determinants of innate immunity and inflammatory responses (34). Moreover, mitochondrial proteins, mtDNA or RNA released into the cytoplasm, for example, because of mitochondrial dysfunction, could act as a pro-inflammatory signal. The most activated upstream regulators in AOX versus WT skeletal muscle are immune-related, notably transcription factors TCF 3 and 4 playing a role in lymphopoiesis, and myeloid differentiation primary response factor 88 (MyD88), involved in NFκB activation (Fig 3C) (Supplemental Data 2). Overall, inflammatory and immune-related pathways, such as STAT3 and interferon signaling and phagocytosis, were down-regulated in the skeletal muscle of AOX mice (Fig 3E). Other highly activated upstream regulators include dysferlin (DYSF), which is thought to be involved in muscle fiber repair (Fig 3C) (Supplemental Data 2).

### AOX expression in mutator mice leads to activation of inflammatory pathways and ISRmt

In contrast to AOX mice, multiple immune-related pathways, including natural killer (NK) cells and IL-8 signaling and phagocytosis, were predicted to be up-regulated in mutator skeletal muscle (Fig 3G). Furthermore, when AOX was also present, pro-inflammatory signaling was even more highly stimulated (Fig 4C), notably IL-1β-driven acute phase response signaling (35) (Fig 4B, C, and E). Compared with mutators, top upstream regulators in AOX–mutators included inflammatory cytokines (IL1β, TNF) and TP53, a major tumor suppressor (Fig 4B, Table S2).

When comparing AOX–mutator with the other mouse groups, molecules involved in signaling of the ISRmt, namely MTHFD2, FGF21, PSAT1, PHGDH, and GDF15, a metabokine that mediates systemic ISRmt signaling alongside FGF21, were up-regulated (Fig 4A and D, Table S1). At the regulator level, in AOX–mutators compared with mutators, Ingenuity Pathway Analysis showed marked activation of ATF-regulated genes and the ISRmt-inducing uncoupling protein 1 (36), which dissipates heat by uncoupling the mitochondrial proton gradient from respiration (Fig 4B, Table S2). In contrast, ISRmt activation seemed to be independent of mTOR signaling, a known regulator (37) (Fig 4C), and of the newly described OMA1–DELE1–HRI regulatory pathway (38) (Supplemental Data 1). Furthermore, the unfolded protein response and folate signaling—both components of ISRmt—were activated (Fig 4C). Upon AOX expression, a similar increase in FGF21 and GDF15 signaling, one-carbon signaling, and UPRmt independent of mTOR signaling was seen previously in a mouse model of mitochondrial myopathy, highlighting potential risks of interfering with ROS signaling in mitochondrial myopathies (22).

Although AOX expression has previously been thought to be mostly innocuous (14, 16, 18, 19, 20, 21, 23), we show here that it induces stress and inflammatory responses in a postmitotic tissue experiencing a broad and progressive mitochondrial RC dysfunction, involving activation of the previously characterized ISRmt (37). Because AOX appears to attenuate rather than enhance ROS signaling (19, 22), our findings indicate that ISRmt induction can occur independently of a ROS signal.

The co-induction of ISRmt and inflammatory signaling in AOX–mutator muscle may be independent or linked. Phosphorylation of eIF2α in response to various physiological stresses, including infection, is an initiator of both processes (39, 40, 41). However, AOX limits an IL-1β-dependent response in bone marrow-derived macrophages (24), suggesting that its effect(s) on inflammation may be tissue-specific. Further research is needed into the significance of ISRmt induction in a setting of mitochondrial dysfunction.

### AOX expression or the Polg mutator alters innate immune signaling

Given the up-regulation of multiple pro-inflammatory mediators in AOX–mutator mice, we examined innate immunity-related genes in our muscle RNA-seq data. Multiple regulators of innate immunity were up-regulated in mutators versus WT mice (Table S2), including IRF3, a critical player in the cGAS–STING

---

(F) Mut, and (H) AOX–mut versus WT in RNA-seq data of mouse skeletal muscle shown as volcano plots. **(E, G, I)** Comparison of the most significant canonical pathways changed in (E) AOX, (G) Mut, (I) AOX-Mut versus WT based on RNA-seq data of mouse skeletal muscle analyzed using ingenuity pathway analysis. WT (n = 5), AOX (n = 5), mutator (n = 5), and AOX–mutator (n = 3), biological replicates, RNA-seq for each sample was performed once. Abbreviations: WT, wildtype mice; AOX, AOX mice; AOX–Mut, AOX–mutator mice; Mut, mutator mice.

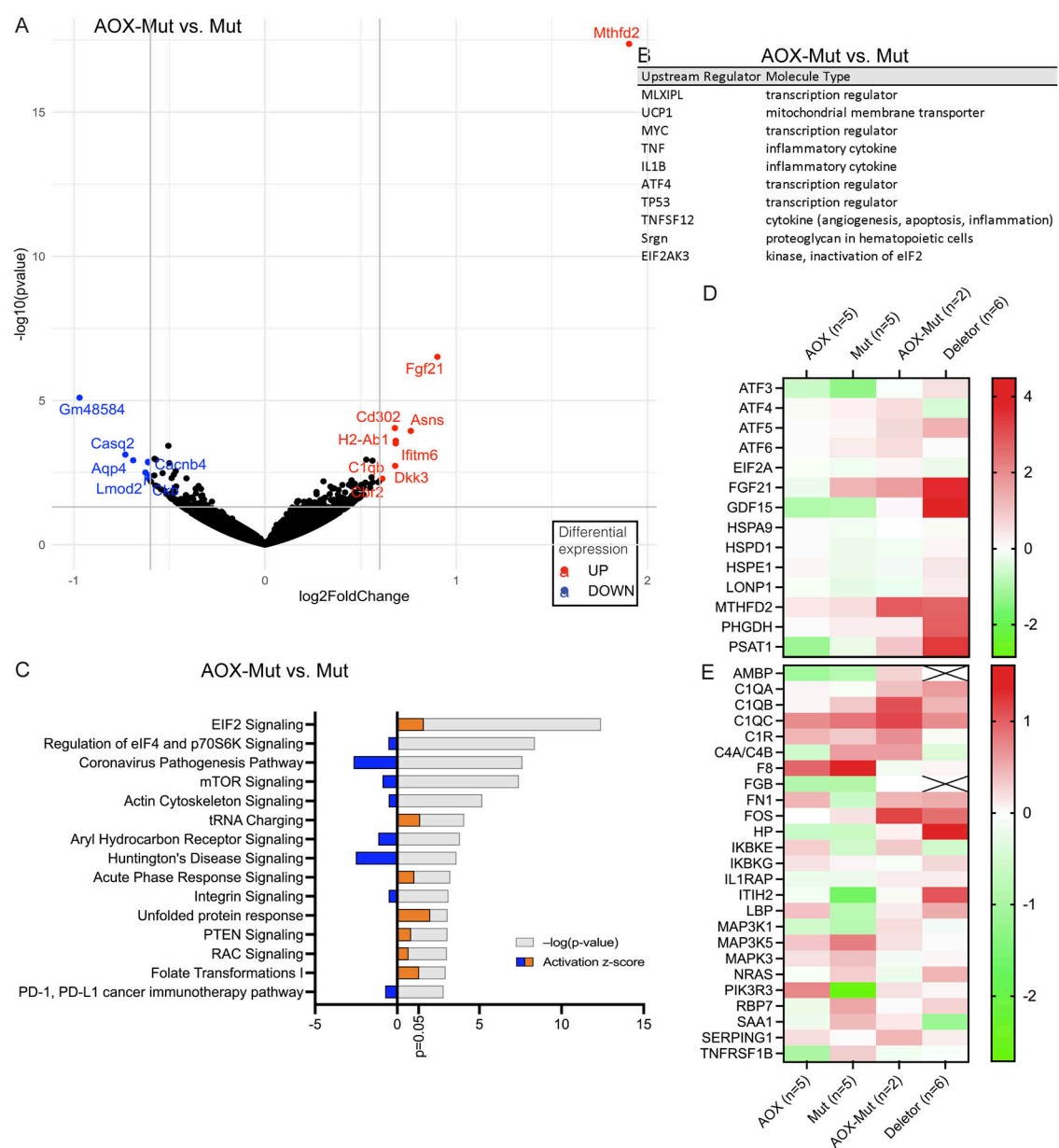

**Figure 4. Alternative oxidase (AOX) expression on a mutator background leads to activation of the mitochondrial integrated stress response (ISRmt) and inflammatory pathways.**

**(A)** Transcripts with the highest experimental fold change differing significantly in AOX–Mut versus Mut in RNA sequencing (RNA-seq) data of mouse skeletal muscle shown as a volcano plot. **(B)** Top 10 activated upstream regulators in AOX–Mut compared with Mut in RNA-seq data of mouse skeletal muscle analyzed using Ingenuity Pathway Analysis (IPA). **(C)** Comparison of the most significant canonical pathways changed in AOX–Mut versus Mut based on RNA-seq data of mouse skeletal muscle analyzed using IPA. **(D, E)** Heatmaps of RNA-seq data of mouse skeletal muscle analyzed using IPA showing (D) mitochondrial integrated stress response genes and (E) acute phase response genes. WT (n = 5), AOX (n = 5), mutator (n = 5), and AOX–mutator (n = 3), biological replicates, RNA-seq for each sample was performed once. Abbreviations: WT, wildtype mice; AOX, AOX mice; AOX–Mut, AOX–mutator mice; Mut, mutator mice.

pathway (42), which is triggered upon cytosolic sensing of mtDNA (43, 44). This has been suggested to contribute to pathologies involving mitochondrial dysfunction (45, 46, 47, 48, 49, 50) and led us to look at the cGAS–STING pathway in more detail using our RNA-seq data.

cGAS and STING, and the cGAS–STING–TBK1–IRF3 signaling cascade, were up-regulated by the expression of AOX and/or the mutator phenotype (Fig 5A and B), and this was true also for

deleter mice, a model for mitochondrial myopathy carrying a dominant patient-equivalent mutation in the mitochondrial replicative helicase TWINKLE (51).

Aberrant activation of the type-I IFN response may aggravate the mutator phenotype (52). This has also been shown in other cases of mitochondrial dysfunction, such as in TFAM (transcription factor A, mitochondrial) heterozygous knockout (*Tfam*$^{+/-}$) mice (48) and in mice with mtDNA stress induced by exhaustive exercise or mtDNA

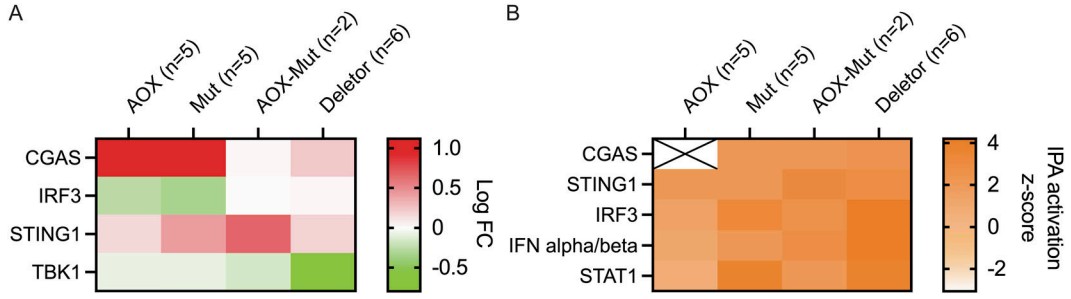

**Figure 5.  Alternative oxidase alters activation of innate immunity pathways in Mutators.**
**(A, B)** Transcriptome analysis; effect of alternative oxidase on skeletal muscle. **(A, B)** Heatmaps of cGAS–STING pathway genes; RNA sequencing data, mouse skeletal muscle; analysis by Ingenuity Pathway Analysis showing comparison based on (A) experimental logarithmic fold change and (B) predicted activation state (z-score). Samples are biological replicates in the numbers presented in the figure; RNA sequencing for each sample was performed once.

mutations in the absence of parkin or PINK1 (47). Activation of innate immunity pathways may therefore be a common response to mitochondrial stress.

Despite much research, no specific therapy is available for pathological oxidative phosphorylation defects. AOX expression has been proposed as a tool not only to study the mechanisms of respiratory chain dysfunction but also to alleviate its pathological consequences (15, 18). The present findings indicate that implementation of AOX, in combination with at least one type of mitochondrial impairment, namely the lifetime accumulation of mtDNA point mutations, activates various stress and immune-related pathways in a tissue-specific fashion. These may be beneficial, as is the case for alleviating mutator-driven anemia, or potentially harmful, as for inflammatory responses in skeletal muscle. Thus, to have any use in therapy, the obvious ethical problems associated with the use of genetic therapies in humans will need to be considered and the effects of AOX expression studied in different tissues and in response to different mitochondrial defects. More broadly, the nature of the link between mitochondrial dysfunction, the stress responses it induces, and altered innate immune signaling warrants broader investigation.

# Materials and Methods

### Mouse models

AOX mice were generated and characterized by Szibor et al (17). Mutator mice (4, 5), also extensively characterized, were generously supplied by Tomas Prolla. Mutator mice are also commercially available (IMSR_JAX:017341; The Jackson Laboratory). Hemizygous AOX mice and heterozygous mutator mice were first crossed. The resultant AOX hemizygous—mutator heterozygous females were further crossed with (a) heterozygous mutator males or (b) homozygous mutator males. Using method (a), the second breeding step resulted in AOX hemizygous—mutator heterozygous mice (5.6%, not used in the study) and heterozygous mutator mice (22.2%, not used in the study), and the study groups used: AOX (27.8%), mutator (22.2%), AOX-mutator double-transgenic (16.7%), and WT littermate mice (27.8%), close to expected Mendelian ratios (16.7% for each group). Using method (b), the second breeding step resulted in AOX hemizygous—mutator heterozygous mice (20.0%, not used in the study) and heterozygous mutator mice

(13.3%, not used in the study) and the study groups AOX–mutator double-transgenic (23.3%) and mutator mice (23.3%) in approximately Mendelian rations (25% for each group); age-matched WT and AOX pups were used (Fig S1). Using methods (a) and (b), 46 mice were obtained. An additional 13 mice were used for NSC extraction. Mice of both sexes were used in the study. No animals were excluded from the analyses. Randomization and blinding were not applicable in this study. The National Animal Experiment Board of Finland approved animal maintenance and experimentation (permit ESAVI-689-04.10.07-2015), and the mice were maintained and studied according to 3R principles. ARRIVE guidelines were followed as applicable.

### NSC

NSC extraction from WT (n = 4), AOX (n = 2), mutator (n = 4), and AOX–mutator (n = 3) mice embryos was performed from the lateral ventricular wall of E11.5–E15.5 mouse brains as previously described (53). Neurospheres were cultured in serum-free Ham's F12 medium (N4888; Sigma-Aldrich) supplemented with B27 (12587010; Gibco), GlutaMAXTM (35050061; Gibco), penicillin–streptomycin (15070073; Gibco), FGF (F0291; Sigma-Aldrich), and EGF (354052; BD). Cell cultures were tested monthly for mycoplasma contamination. Analysis of NSC self-renewal capacity was performed as previously described (53). To determine the proliferation rate, a BrdU incorporation assay was used; neurospheres were incubated in 10 $\mu$M BrdU (BD PharMingen), stained with anti-BrdU and fluorescent secondary antibody, and analyzed using a FACSAria Cell Sorter.

### Western blot

Whole-cell protein extraction from NSCs was performed as in reference 8. Protein concentration was measured using the Bradford method (Protein Assay; Bio-Rad). For SDS–PAGE before Western blotting, sample aliquots were mixed with 3 × SDS loading buffer, denatured, and resolved on 4–20% Mini-PROTEAN TGX Stain-Free Gels (#4568096; Bio-Rad) at 100–120 V for 1 h. Proteins were transferred to PVDF membranes using a Trans-Blot Turbo RTA Mini transfer kit (#1704727; Bio-Rad). Membranes were blocked with 5% milk in 1x TBST for 1 h at RT, then probed overnight at 4°C with a custom-manufactured anti-AOX antibody (1:33,000, (54); polyclonal

rabbit serum) in 5% milk, then washed three times in a blocking solution. Incubation with HRP-conjugated goat anti-rabbit secondary antibody (111-035-144, 1:10,000; Jackson ImmunoResearch) was for 1 h at RT, followed by three washes in TBST. Imaging was done using a ChemiDoc imaging system (Bio-Rad).

### Blood analyses

Blood was collected from euthanized mice into EDTA tubes (Greiner Bio-One) by heart puncture. A blood count was performed from whole-blood samples on the same day using an Advia 2120i analyzer (Siemens).

### FACS

Euthanized mice's femoral and tibial bones were cut out and cleaned of soft tissue. Bone marrow was flushed out using a syringe and 1 ml of PBS + 5% FBS and filtered through a 40-$\mu$m strainer (Dako). Cells were counted and analyzed immediately. Hematopoietic lineages of adult bone marrows were analyzed using a BD Influx Cell Sorter (Beckton Dickinson). CD16/CD32 blocker (1 $\mu$g/1 × $10^6$ cells; BD Pharmingen) was used to inhibit possible nonspecific binding of the antibodies. Fluorescence-conjugated antibodies against CD71 (BD Pharmingen), CD11b (BD Pharmingen), Ter119 (eBioscience), and B220 (BD Pharmingen) were used (each 1 $\mu$g/1 × $10^6$ cells) for staining. Propidium iodide was used as a dead cell marker. Unstained and fluorescence-minus-one controls were used to determine the autofluorescence of the cells and gates for the cell populations of interest, respectively. One hundred thousand cells per sample were analyzed.

### RNA extraction

Total RNA was extracted from skeletal muscle (*m. quadriceps femoris*) using TRIzol reagent (Invitrogen) and purified using RNeasy Mini Kit (QIAGEN). The samples were homogenized using TRIzol reagent (Invitrogen), and chloroform was added to allow homogenate separation. The lysate–ethanol mix was then purified using RNeasy Mini Kit (QIAGEN) according to the manufacturer's instructions. Extracted RNA was treated with RNase-free DNAse (M6101; Promega). 1,000 ng of total RNA was used to generate cDNA using a MAXIMA cDNA synthesis kit (K1641; Thermo Fisher Scientific).

### Transcriptomics analysis

RNA from WT (n = 5), AOX (n = 5), mutator (n = 5), and AOX–mutator (n = 3) was submitted for transcriptomic analysis. RNA quality-control analysis was done using Tapestation 4200 (Agilent). RNA-seq was performed using a "Bulkseq" 3'UTR-counting gene expression profiling method based broadly on BRBseq/Dropseq with high-output (1 × 75 bp) read lengths. The Biomedicum Functional Genomics Unit provided the service at the Helsinki Institute of Life Science and Biocenter Finland at the University of Helsinki. Primary data analysis was done using the DeSeq2 package from Bioconductor release 3.9 (55) in R Studio version 1.4.1103 using R version 3.6.3. Data were

analyzed further through Ingenuity Pathway Analysis (QIAGEN Inc., https://digitalinsights.qiagen.com/products-overview/discovery-insights-portfolio/analysis-and-visualization/qiagen-ipa/) (56).

### Histology

Skeletal muscle (*m. quadriceps femoris*, *QF*), heart muscle, and brain were collected from WT, AOX, mutator, and AOX–mutator mice aged 40 wk. The tissues were harvested immediately after sacrificing the mice, embedded in OCT Compound embedding medium (Tissue-Tek), and snap-frozen in a 2-methylbutane bath in liquid nitrogen. In situ histochemical COX and SDH activities were analyzed from frozen tissue sections (12 $\mu$m) using standard protocols (51). Imaging was done by light microscopy (Axioplan 2 Universal Microscope; Zeiss). Approximately 200–700 fibers from each mouse in the study groups WT (n = 6), AOX (n = 8), mutator (n = 8), and AOX–mutator (n = 4) were counted to calculate the percentage of COX-negative and COX-negative/SDH-positive fibers from QF sections. The COX-negative and SDH-positive fibers were quantified from each study group using ImageJ software (57).

Immunofluorescent staining of SDHA (complex II), COX-I (complex IV), nuclei, and laminin in the muscle of WT (n = 3), AOX (n = 3), mutator (n = 4), and AOX–mutator (n = 2) mice was done by adapting a previously published protocol (58). Primary antibodies used were anti-SDHA mouse IgG1 (ab14715, 1:95; Abcam), anti-MTCO1 mouse IgG2a (ab14705, 1:95; Abcam), and anti-laminin (L9393, 1:100; Sigma-Aldrich). Secondary antibodies used were goat anti-mouse IgG1 biotin (ab97238, 1:200; Abcam) with streptavidin conjugate Alexa Fluor 647 (S21374, 1:100; Invitrogen), goat anti-mouse IgG2a Alexa Fluor 488 (A-21136, 1:200; Invitrogen), and goat anti-rabbit Alexa Fluor 568 (A-11011, 1:100; Invitrogen), respectively. Slides were mounted with ProLong Diamond Antifade Mountant with DAPI (P36966; Invitrogen). Images were generated using a 3DHISTECH Pannoramic 250 FLASH II digital slide scanner at the Genome Biology Unit supported by HiLIFE and the Faculty of Medicine, University of Helsinki, and Biocenter Finland. Digitalized images were visualized using 3DHISTECH CaseViewer 2.2.0 software. SDHA and COX-I staining intensity and fiber size were quantified using CellProfiler (59, 60).

Additional immunofluorescent staining of COX-I and TOM20 was done on muscle of WT (n = 3), AOX (n = 3), Mutator (n = 4), and AOX-Mutator (n = 1) mice. COX-I was stained as above. For TOM20, the primary antibody used was anti-TOMM20 rabbit (ab186734; Abcam) and the secondary antibody was chicken anti-rabbit IgG (H+L) Alexa Fluor 594 (A-21442; Invitrogen). Slides were mounted with Fluoromount (F4680-25ML; Sigma-Aldrich). Imaging was done by light microscopy (Zeiss Axioimager Z2; ZEISS) at 20X magnification. Colocalization of COX-I and TOM20 was quantified using CellProfiler (59, 60).

MHC isoforms were stained in skeletal muscle (*m. quadriceps femoris*) of WT (n = 5), AOX (n = 7), mutator (n = 8), and AOX–mutator (n = 5) mice using a protocol adapted from previous reports (61, 62). Fresh frozen 10-$\mu$m muscle sections were used. Primary antibodies were obtained from the Developmental Studies Hybridoma Bank at the University of Iowa: MyHC I antibody BA-D5 1:40 (63), MyHC IIa antibody SC-71 1:200 (63), MyHC IIb antibody BF-F3 1:100 (64), and MyHC IIx antibody 6H1 1:50 (65). Secondary antibodies used were

goat anti-mouse IgG2a Alexa Fluor 488 (A-21131, 1:200; Invitrogen) with BA-D5, goat anti-mouse IgG (H+L) Alexa Fluor 488 (A-11001, 1:200; Invitrogen) with SC-71, and goat anti-mouse IgM Alexa Fluor 488 (A-21042, 1:100; Invitrogen) with BF-F3 and 6H1. In addition, slides were co-stained for laminin (primary antibody: L9393, batch 82508, 1:100; Sigma-Aldrich) with secondary antibody goat anti-rabbit IgG (H+L) Alexa Fluor 594 (R37119, 1:300; Invitrogen). Slides were mounted with Vectashield Antifade Mounting Medium with DAPI (H-1200-10; Vector). Images were taken using the Axio Imager M1 microscope (ZEISSAXIOM1; ZEISS). Quantification of positive fibers for each MyHC type was done using ImageJ software ([57]).

AOX was stained in in skeletal muscle (*m. quadriceps femoris*) of WT (n = 3), AOX (n = 3), mutator (n = 4), and AOX–mutator (n = 1) mice. Fresh frozen 10-$\mu$m muscle sections were used. The primary AOX antibody (([66]); 21st Century Biochemicals) was a kind donation from Eric Dufour. Goat anti-rabbit IgG (H+L) Alexa Fluor 647 (A-21245, 1:300; Invitrogen) was used as the secondary antibody. Slides were mounted with Fluoromount (F4680-25ML; Sigma-Aldrich). Imaging was done by light microscopy (Zeiss Axioimager Z2; ZEISS) at 20X magnification. At least 500 fibers from each mouse in the study groups were measured for intensity using ImageJ ([57]). Background subtraction was done by rolling ball with sliding paraboloid at 50 pixels before quantification of mean gray values in ImageJ.

## Statistical analyses

One-way ANOVA for groups displaying normal distribution followed by Tukey's multiple comparisons test was performed on independent observations using GraphPad Prism version 9.0.2 for Mac, GraphPad Software, www.graphpad.com. *P*-values are shown as ns ($P > 0.05$), * ($P \leq 0.05$), ** ($P \leq 0.01$), *** ($P \leq 0.001$), and **** ($P \leq 0.0001$). Graphs show the mean with SD with individual measurements shown.

## Data Availability

All original data are available from the authors upon request. The RNA-sequencing data from this publication have been deposited to the NCBI Sequence Read Archive (SRA) database and assigned the BioProject ID PRJNA1006295.

## Supplementary Information

## Acknowledgements

We are grateful to Markus Innilä, Babette Hollman, Sonja Jansson, Maarit Partanen, and Tea Tuomela for technical assistance; Mito Takayuki, Christopher Jackson, and Nahid Khan for assistance with imaging; Gulayse Ince-Dunn for RNAseq insight; Yilin Kang for helpful discussions; and Biomedicum Imaging Unit and Biomedicum Functional Genomics Unit for services and infrastructure. The authors acknowledge funding from the Biomedicum Helsinki Foundation (L Ikonen), The Finnish Medical Foundation (L Ikonen), the Academy of Finland (A Suomalainen, HT Jacobs, and M Szibor), the Sigrid Jusélius Foundation (L Ikonen and A Suomalainen), the University of Helsinki (A Suomalainen), and the European Research Council (HT Jacobs and M Szibor). The work used core facilities part-supported by funding from Biocenter Finland.

## Author Contributions

L Ikonen: data curation, funding acquisition, investigation, visualization, and writing—original draft, review, and editing.
S Pirnes-Karhu: conceptualization, investigation, and writing—original draft, review, and editing.
S Pradhan: investigation and visualization.
HT Jacobs: resources, funding acquisition, methodology, and writing—review and editing.
M Szibor: resources, funding acquisition, methodology, and writing—review and editing.
A Suomalainen: conceptualization, resources, data curation, supervision, funding acquisition, project administration, and writing—original draft, review, and editing.

## Conflict of Interest Statement

M Szibor is a shareholder in a startup company aiming at developing therapeutics based on AOX. All other authors declare no competing interests.

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
