## [Reviewer comments · Life Science Alliance]

Life Science Alliance

Alternative oxidase causes cell-type- and tissue-specific responses in mutator mice

Lilli Ikonen, Sini Pirnes-Karhu, Swagat Pradhan, Howard Jacobs, Marten Szibor, and Anu Suomalainen

DOI: <https://doi.org/10.26508/lsa.202302036>

Corresponding author(s): Lilli Ikonen, University of Helsinki and Anu Suomalainen, University of Helsinki

Review Timeline:

Submission Date:	2023-03-14
Editorial Decision:	2023-04-11
Revision Received:	2023-07-11
Editorial Decision:	2023-07-31
Revision Received:	2023-08-17
Accepted:	2023-08-18

Scientific Editor: Novella Guidi

Transaction Report:

April 11, 2023

Re: Life Science Alliance manuscript #LSA-2023-02036-T

Lilli P Pihlajamäki
University of Helsinki
Stem Cells and Metabolism Research Program, Faculty of Medicine
Biomedicum 1
Haartmaninkatu 8
Helsinki 00290
Finland

Dear Dr. Pihlajamäki,

Thank you for submitting your manuscript entitled "Alternative oxidase expression in mtDNA mutator mice improves blood phenotype but enhances inflammatory and stress responses in skeletal muscle" to Life Science Alliance. The manuscript was assessed by expert reviewers, whose comments are appended to this letter. We invite you to submit a revised manuscript addressing the Reviewer comments.

Thank you for this interesting contribution to Life Science Alliance. We are looking forward to receiving your revised manuscript.

Sincerely,

B. MANUSCRIPT ORGANIZATION AND FORMATTING:

Reviewer #1 (Comments to the Authors (Required)):

This research provides important information regarding the possibility of using alternative oxidase (AOX) as a therapeutic option in primary mitochondrial diseases. To this end, authors use a unique mutator-AOX model while examining tissue specific changes with the emphasis on hematopoiesis, inflammation and stress response in bone marrow muscle and brain. The research is thorough and the results are interesting. I have only a few comments as follows;

- 1- In the introduction/discussion please relate also to AOX in another mouse mitochondrial cardiomyopathy model PMID30530468
- 2- Explain the meaning of Ter119
- 3- Was there any difference between skeletal and heart muscle/was there any cardiomyopathy?
- 4- Were any functional studies performed i.e muscle strength
- 5- Why only muscle specifically chosen for RNA-seq and not bone marrow which showed improvement with AOX?
- 6- Why is the breeding plan in fig s4? should it not be s1?
- 7- Could anti-inflammatory treatment such as NSAIDs be helpful in ameliorating the negative impact of AOX in muscle?

Reviewer #2 (Comments to the Authors (Required)):

Pihlajamäki and colleagues describe here the effect of the expression of the AOX from *Ciona intestinalis* on the blood and neural stem cell pools of the mutator mouse. The authors found that AOX improved the anemia by ameliorating the erythrocytes differentiation. In contrast, it did not affect the stemness of neural stem cells. The authors then moved to the analysis of the therapeutic potential of AOX examining the respiratory chain in the skeletal muscle of mutator/AOX mice. No COX deficient/SDH positive fibers were detected in the mutator mice, but a general reduction of cox staining, with some fibers staining particularly intense. The interpretation of the authors is that AOX dysregulates the respiratory chain. AOX however did not affect the fiber type compared to naïve mutator mice. Also, other phenotypes of mutator mice in the brain were not changed by AOX. Interestingly and unexpectedly, AOX induced transcriptional and epigenetics changes in skeletal muscle compared to WT mice. In addition, the upregulation of inflammatory pathways and in the mitochondrial integrated stress response was observed. Overall, these results are in contrast to previous reports which suggested no gross effects of AOX in wild-type animals. The paper is potentially interesting, but the authors should clarify some relevant issues.

Main point.

The authors do not show in the paper that AOX is active in their samples. AOX has been reported to be engaged in the respiratory chain only when coenzyme Q is reduced over a certain threshold. However, I am not convinced this happens with an overall mild reduction of COX, which is only shown by histochemical analysis. I suggest the authors quantitatively measure COX and SDH activities in the mutator/mutator-AOX mice in the absence of a specific assay for AOX.

Additional points.

First, the expression of AOX in NPCs can be easily assessed by immunofluorescence, and this would clarify if the different effects of AOX in erythrocyte precursors and NPCs can be due to different AOX expression. Second, the authors report the presence of cells with particularly strong staining in the skeletal muscle, which has never been reported before. Do these cells also have more intense SDH staining? This would confirm increased mitochondrial biogenesis.

Reviewer #3 (Comments to the Authors (Required)):

This paper from Pihlajamäki and colleagues reports the detailed assessment of wt and mutator mice that express the alternative oxidase AOX. It is a very interesting and well written paper with a lot of intriguing data. Much of the data is challenging to interpret but highlights the complications of tissue specificity in mitochondrial disease. The most clear positive of AOX

expression is its role in ameliorating the defect of mutator (mut) in haematopoiesis. This at first appears to suggest that AOX can have positive effects by reducing ROS levels but this simplistic assessment does not hold for many other tissues. AOX expression on its own seems to result in upregulation of markers involved in innate immunity and when expressed in the mut background there is an increase in several aspects of the integrated stress response. It would be great to know what is causing this as a function of AOX expression but it is not necessary for publication at this stage.

The work is well performed and the data is of interest. I have a couple of points the authors may wish to consider:

1. On p6, please briefly explain what Ter119 is being used as a marker for.
2. Gross phenotype of the AOX-mut and mut were similar, with progeria onset at similar stage. The data shows nicely that there is a positive effect of AOX expression on the problems of haematopoiesis demonstrated by the mut. The authors say this is promising, as it appears to be the anaemia that is ultimately responsible for the early death of the muts. The obvious question is whether the AOX expression increases the life expectancy of the muts. If not, it would seem that the defect in haematopoiesis is perhaps not the cause of early death ?
3. The inference is made that low level overall staining of COX1 means a decrease in the level of mitochondrial protein in the muscle fibres. The use of a marker of mito mass such as VDAC or TOM20 would have been useful to support this inference.
4. The para at the end of p.10, 'The number of transcripts.....' I found very difficult to follow. Could this be reworded for clarity ?

Dear Dr. Novella Guidi and *Life Science Alliance* editorial team,

Thank you for giving us the opportunity to submit a revised version of our manuscript "Alternative oxidase expression in mtDNA mutator mice improves blood phenotype but enhances inflammatory and stress responses in skeletal muscle" to *Life Science Alliance*. We appreciate the time the reviewers have taken to provide valuable insight and comments on our manuscript.

Changes to the manuscript are highlighted in bold font.

RNA sequencing data are to be uploaded to Genbank.

Here is a point-by-point response to the reviewers' comments and concerns.

Comments from Reviewer 1

- 1) In the introduction/discussion please relate also to AOX in another mouse mitochondrial cardiomyopathy model PMID30530468.

Answer: Thank you for pointing this out. We have added this to the introduction.

- 2) Explain the meaning of Ter119.

Answer: Thank you for noting this lacking explanation, which has now been added to the text.

- 3) Was there any difference between skeletal and heart muscle/was there any cardiomyopathy?

Answer: No. We did not see differences in muscle fiber size between groups in skeletal or heart muscle. Further, we did not observe evidence of either hypertrophic or dilating cardiomyopathy as analyzed by echocardiography. Because we did not observe any significant differences here, further analysis was not carried out. We have now mentioned this in the manuscript on page 8.

- 4) Were any functional studies performed i.e muscle strength?

Answer: Thank you for the question. Mutator mice are frail because of many affected systems leading to anemia and osteoporosis among other phenotypes. Therefore, we focused on objective molecular markers.

- 5) Why only muscle specifically chosen for RNA-seq and not bone marrow which showed improvement with AOX?

Answer: Thank you for this important question. Bone marrow is greatly affected in mutators leading to very different amounts of both erythroid and lymphoblastoid lineage cells in the bone marrow of mutators and wild-type mice. Therefore, relevant analysis of bone marrow would only have been possible by single-cell RNA sequencing, which was not in the scope of this study.

6) Why is the breeding plan in fig s4? should it not be s1?

Answer: This is an excellent point. The figure has been moved.

7) Could anti-inflammatory treatment such as NSAIDs be helpful in ameliorating the negative impact of AOX in muscle?

Answer: Thank you for the insightful question. Originally, we were interested in the impact of AOX on stem cell phenotypes of mutators because ROS is an important factor for stem cell differentiation, and then we expanded the question to muscle. This study is a proof-of-principle of the effects of AOX on different cell types and in a disease vs WT context. Although an interesting question, we did not add further components to the set-up – the topic was out of the scope of the paper.

Comments from Reviewer 2

Main point. The authors do not show in the paper that AOX is active in their samples. AOX has been reported to be engaged in the respiratory chain only when coenzyme Q is reduced over a certain threshold. However, I am not convinced this happens with an overall mild reduction of COX, which is only shown by histochemical analysis. I suggest the authors quantitatively measure COX and SDH activities in the mutator/mutator-AOX mice in the absence of a specific assay for AOX.

Answer: Thank you for giving us the opportunity to answer this point. The engagement of AOX mainly depends on the balance of substrate use making it unlikely that the observed effects would primarily be consequences of protein inactivity. To study AOX engagement it would be necessary to also test extensively for possible interacting factors including diet, circadian cycles, and hormonal signaling. This would be an interesting study in its own right but is beyond the scope of the present work. We consider the physiological effects that are both cell-type and disease-stage dependent to be strong evidence of the activity of AOX.

In addition to AOX expression, we have confirmed the presence of AOX protein in skeletal muscle by immunofluorescence. These new data are included in the Figure S2B–C.

Additional points.

1) First, the expression of AOX in NPCs can be easily assessed by immunofluorescence, and this would clarify if the different effects of AOX in erythrocyte precursors and NPCs can be due to different AOX expression.

Answer: We would like to point out that in the Figure S1A in the original manuscript (now S2A) the AOX expression levels in the neural stem cells of AOX and AOX-mutator mice were reported.

2) Second, the authors report the presence of cells with particularly strong staining in the skeletal muscle, which has never been reported before. Do these cells also

have more intense SDH staining? This would confirm increased mitochondrial biogenesis.

Answer: Thank you for the question. These hyperpositive fibers are present in COX/SDH histochemical activity staining on frozen muscle sections, but not when detecting the protein amounts by immunofluorescence. The data indicate that the fibers have upregulated COX activity but not increased in protein amount. Actually, in the original manuscript Figure S3B–C (now Figure S4B–C), we assessed the amount of SDHA protein by immunofluorescence from the same muscle sections as COX1 amount. By mistake the result was only shown in the figure, and not mentioned in the text, and now has been added also to the text on page 9.

Comments from Reviewer 3

This paper from Pihlajamäki and colleagues reports the detailed assessment of wt and mutator mice that express the alternative oxidase AOX. It is a very interesting and well written paper with a lot of intriguing data. Much of the data is challenging to interpret but highlights the complications of tissue specificity in mitochondrial disease. The most clear positive of AOX expression is its role in ameliorating the defect of mutator (mut) in haematopoiesis. This at first appears to suggest that AOX can have positive effects by reducing ROS levels but this simplistic assessment does not hold for many other tissues. AOX expression on its own seems to result in upregulation of markers involved in innate immunity and when expressed in the mut background there is an increase in several aspects of the integrated stress response. It would be great to know what is causing this as a function of AOX expression but it is not necessary for publication at this stage.

Answer: We thank for the comment and agree.

- 1) On p6, please briefly explain what Ter119 is being used as a marker for.

Answer: Thank you for pointing this out. Ter119 is a glycoprotein A-associated protein which marks erythroid lineage cells separate from other blood cells, from proerythroblasts to mature erythrocytes. We have added this to the introduction.

- 2) Gross phenotype of the AOX-mut and mut were similar, with progeria onset at similar stage. The data shows nicely that there is a positive effect of AOX expression on the problems of haematopoiesis demonstrated by the mut. The authors say this is promising, as it appears to be the anaemia that is ultimately responsible for the early death of the muts. The obvious question is whether the AOX expression increases the life expectancy of the muts. If not, it would seem that the defect in haematopoiesis is perhaps not the cause of early death?

Answer: This is an excellent point. If mutators are allowed to age, their hemoglobin levels decrease down to 50 g/l which is life-limiting before any other of their symptoms are critical. While their hematopoiesis is somewhat improved, Hb still decreases in AOX-mutators. Our ethical licenses do not allow us to keep these mice alive for a true lifespan experiment.

- 3) The inference is made that low level overall staining of COX1 means a decrease in the level of mitochondrial protein in the muscle fibres. The use of a marker of mito mass such as VDAC or TOM20 would have been useful to support this inference.

Answer: To answer this question, we did further immunofluorescent co-staining of TOM20, which showed no major mitochondrial mass change except for in AOX-WT mice that appeared to have slightly lowered Tom20 and SDH protein (Figure S4B–D).

- 4) The para at the end of p.10, 'The number of transcripts.....' I found very difficult to follow. Could this be reworded for clarity ?

Answer: Thank you for pointing this out. This paragraph has now been rewritten to improve clarity.

References

1. Ahlqvist KJ, Hämäläinen RH, Yatsuga S, Uutela M, Terzioglu M, Götz A, et al. Somatic progenitor cell vulnerability to mitochondrial DNA mutagenesis underlies progeroid phenotypes in Polg mutator mice. *Cell Metab.* 2012;15(1):100-9.
2. Manish B, Partho PS. How to interpret an echocardiography report (for the non-imager)? *Heart.* 2017;103(21):1733.
3. Tugendreich S, Pearson CI, Sagartz J, Jarnagin K, Kolaja K. NSAID-Induced Acute Phase Response is Due to Increased Intestinal Permeability and Characterized by Early and Consistent Alterations in Hepatic Gene Expression. *Toxicologic Pathology.* 2006;34(2):168-79.
4. Tsutsumi S, Gotoh T, Tomisato W, Mima S, Hoshino T, Hwang HJ, et al. Endoplasmic reticulum stress response is involved in nonsteroidal anti-inflammatory drug-induced apoptosis. *Cell Death & Differentiation.* 2004;11(9):1009-16.
5. Chen JS, Alfajaro MM, Chow RD, Wei J, Filler RB, Eisenbarth SC, et al. Non-steroidal anti-inflammatory drugs dampen the cytokine and antibody response to SARS-CoV-2 infection. *J Virol.* 2021;95(7).
6. Brandolini L, Antonosante A, Giorgio C, Bagnasco M, d'Angelo M, Castelli V, et al. NSAIDs-dependent adaption of the mitochondria-proteasome system in immortalized human cardiomyocytes. *Sci Rep.* 2020;10(1):18337.

July 31, 2023

RE: Life Science Alliance Manuscript #LSA-2023-02036-TR

Dr. Lilli P Pihlajamäki
University of Helsinki
Stem Cells and Metabolism Research Program, Faculty of Medicine
Biomedicum 1
Haartmaninkatu 8
Helsinki 00290
Finland

Dear Dr. Pihlajamäki,

Thank you for submitting your revised manuscript entitled "Alternative oxidase causes cell-type- and tissue-specific re-sponses in mutator mice". We would be happy to publish your paper in Life Science Alliance pending final revisions necessary to meet our formatting guidelines.

- please add the Twitter handle of your host institute/organization as well as your own or/and one of the authors in our system
- please note that there is a name discrepancy in the presentation of the name of one of your co-authors; please correct (Anu Suomalainen-Wartiovaara in the system vs. Anu Suomalainen in the manuscript file)
- please consult our manuscript preparation guidelines <https://www.life-science-alliance.org/manuscript-prep> and make sure your manuscript sections are in the correct order
- please add callouts for Figures 3B; S2B-E; S4H to your main manuscript text

A. FINAL FILES:

B. MANUSCRIPT ORGANIZATION AND FORMATTING:

Sincerely,

Reviewer #1 (Comments to the Authors (Required)):

The revised MS is improved and adequately addresses the reviewer's critique. I have no further comments.

Reviewer #2 (Comments to the Authors (Required)):

In this new version, the authors fully addressed my original concerns and I have no other objections/suggestions.

Reviewer #3 (Comments to the Authors (Required)):

I'd like to thank the authors for their response to my comments. I am happy to recommend publication as is.

August 18, 2023

RE: Life Science Alliance Manuscript #LSA-2023-02036-TRR

Dr. Lilli P Ikonen
University of Helsinki
Stem Cells and Metabolism Research Program, Faculty of Medicine
Biomedicum 1
Haartmaninkatu 8
Helsinki 00290
Finland

Dear Dr. Ikonen,

Thank you for submitting your Research Article entitled "Alternative oxidase causes cell-type- and tissue-specific responses in mutator mice". It is a pleasure to let you know that your manuscript is now accepted for publication in Life Science Alliance. Congratulations on this interesting work.

DISTRIBUTION OF MATERIALS:

Again, congratulations on a very nice paper. I hope you found the review process to be constructive and are pleased with how the manuscript was handled editorially. We look forward to future exciting submissions from your lab.

Sincerely,
